# Heartbeat-Evoked Cortical Potential during Sleep and Interoceptive Sensitivity: A Matter of Hypnotizability

**DOI:** 10.3390/brainsci11081089

**Published:** 2021-08-19

**Authors:** Lucia Billeci, Ugo Faraguna, Enrica L. Santarcangelo, Paola d’Ascanio, Maurizio Varanini, Laura Sebastiani

**Affiliations:** 1Institute of Clinical Physiology, National Council of Research (CNR), 56100 Pisa, Italy; lucia.billeci@ifc.cnr.it (L.B.); varanini@ifc.cnr.it (M.V.); 2Department of Translational Research and New Technologies in Medicine and Surgery, University of Pisa, 56127 Pisa, Italy; ugo.faraguna@unipi.it (U.F.); paola.dascanio@unipi.it (P.d.); laura.sebastiani@unipi.it (L.S.); 3Department of Developmental Neuroscience, IRCCS Stella Maris Foundation, 56121 Pisa, Italy

**Keywords:** interoceptive sensitivity, interoceptive accuracy, sleep, cortical potential, hypnotic susceptibility

## Abstract

Individuals with different hypnotizability display different interoceptive sensitivity/awareness (IS) and accuracy (IA), likely sustained by morphofunctional differences in interoception-related brain regions and, thus, possibly also observable during sleep. We investigated the heartbeat-evoked cortical potential amplitude (HEP) during sleep, its association with IS, and the role of hypnotizability in such association. We performed a retrospective analysis of polysomnographic recordings of 39 healthy volunteers. Participants completed the Multidimensional Assessment of Interoceptive Awareness (MAIA), measuring IS and IA, and underwent hypnotic assessment via the Stanford Hypnotic Susceptibility Scale, form A. The amplitude of the early and late HEP components was computed at EEG frontal and central sites. In both regions, the early HEP component was larger in N3 than in N2 and REM, with no difference between N2 and REM. Greater HEP amplitude at frontal than at central sites was found for the late HEP component. HEP amplitudes were not influenced by the autonomic state assessed by heart rate variability in the frequency and time domains. We report for the first time a positive correlation between the central late HEP component and MAIA dimensions, which became non-significant after removing the effects of hypnotizability. Our findings indicate that hypnotizability sustains the correlation between IS and HEP amplitude during sleep.

## 1. Introduction

Interoception represents the experience of the physiological condition of the body and is due to the integration of visceral signals at high levels within the central nervous system [1]—that is, in the insular, anterior cingulate, prefrontal, and somatosensory cortices [2,3,4,5,6,7]. The activity of these areas has been associated with autonomic activities, inducing changes in pupil size, as well as cardiac, respiratory, intestinal, and electrodermal activity [8,9,10]. Variations in grey matter volume have been observed in patients with altered interoception, such as those with borderline personality disorder [11], depersonalization [12], insomnia [13], sleep fragmentation [14], panic disorders [15], and multiple sclerosis [16]. 

Interoception, however, is a multifaceted construct [1], including the ability to detect visceral information (accuracy, IA), as well as the tendency to be aware of it and the modes of its interpretation (IS, awareness/sensitivity). IA has been usually measured by the heartbeat detection test [17], although recent evidence challenges the appropriateness of this test to specifically indicate IA [18]. IS is measured by questionnaires such as the Multidimensional Assessment of Interoceptive Awareness (MAIA) [4,19], which shows the same limitations characterizing every self-reported assessment [20]. 

The activation of brain areas preferentially responding to interoceptive signals has been associated with the MAIA dimensions [12] representing the tendency to ignore sensations of pain/discomfort (*not distracting*) and to not experience emotional distress due to unpleasant sensations *(not worrying*), as well as the ability to focus attention to body sensation *(attention regulation*). Altered IS has been found in patients diagnosed with alexithymia [21], anxiety and somatic symptoms [22], epilepsy [23], anorexia nervosa [24], and impulsivity [25].

### 1.1. Heartbeat-Evoked Cortical Potential 

Interoceptive accuracy (IA) measured by the heartbeat detection test [17] has been positively correlated with the cortical activity related to cardiac afferents—that is, heartbeat-evoked potential (HEP) [26,27]—although the cardiac signal may also contain non-interoceptive information [28]. No significant association has been found, however, between HEP amplitudes and the sympathetic–parasympathetic state [19,26,29].

Greater HEP has been described in patients with depersonalization [12], anorexia nervosa [24], and borderline personality disorder [30] during wakefulness, as well as in patients with nightmares during REM sleep [31]. Greater HEP amplitudes associated with higher IS subjectively reported through MAIA have been observed in individuals with insomnia [29], and a positive correlation of the HEP amplitude with the *not worrying* dimension of interoceptive sensitivity has been shown during wakefulness [32], whereas no information is currently available about the relationship between IS and HEP amplitudes during sleep. 

During sleep, exteroceptive and interoceptive information are processed differentially, as the latter seems to be less vulnerable to the disruption of sensory integration occurring during deep and REM sleep [33,34]. The cortical correlates of exteroceptive stimuli, in fact, are observable during REM only occasionally and when evoked by salient stimuli [35], whereas the HEP amplitude has been found to be greater during deep sleep (N3) than during N2, and no difference has been observed between N2 and REM [36]. 

### 1.2. Interoception and Hypnotizability

Hypnotizability—the proneness to respond to imaginative suggestions—is a psychophysiological trait [37] the remains stable throughout life [36,38,39], and can be measured by a validated scale classifying highly (highs), moderately (mediums), and less hypnotizable individuals (lows). 

With respect to lows, highs display greater proneness to change their bodily state [40,41], i.e., easily entering sleep [42,43,44] and hypnosis [40], experiencing relaxation characterized by greater parasympathetic activity [45,46], and by reduced excitability of spinal motor neurons [47,48] during long-lasting relaxation sessions. Highs also exhibit lower sympathetic activation in the shift from sitting to standing position [49], and less reduced release of endothelial nitric oxide during mental stress and nociceptive stimulation [50,51]. In addition, they exhibit morphofunctional brain characteristics that are theoretically able to influence interoception. Compared to lows, in fact, highs display reduced grey matter volume (GMV) in the insula and in the left cerebellar lobules IV–VI, increased GMV in the mid-temporal and mid-occipital cortices and in the superior frontal gyrus, and stronger functional connectivity between the anterior cingulate cortex and the dorsolateral prefrontal cortex [52,53]. Highs exhibit lower IA measured by heartbeat detection test than do lows [54], and greater IS measured by MAIA [55] with respect to lows and mediums, possibly based on these morphofunctional brain differences. Thus, interoceptive processing may differ according to hypnotizability, and such differences may also appear during sleep.

### 1.3. Aim of the Study

Since highs display greater IS [55] and lower IA during wakefulness [54] with respect to lows/mediums, the aims of the present study were to assess, for the first time, the possible association between IS and HEP amplitude during sleep, and to investigate whether hypnotizability is involved in their relationship. We also replicated the observation of differences in the HEP amplitude between REM and non-REM sleep [32], and confirmed the lack of association between HEP amplitude and heart rate/heart rate variability (HRV) in the entire sample [12,26].

## 2. Methods

### 2.1. Subjects 

The study reports a retrospective analysis of portable PSG (polysomnographic) recordings obtained from 39 healthy volunteers (age (mean ± SD): 23 ± 2.24 years; 26 females) enrolled in an earlier study [56] approved by the Bioethics Committee of the University of Pisa. Exclusion criteria were the anamnesis of medical, neurological, and psychiatric disorders, sleep disturbance, and intake of psychoactive drugs over the previous 6 months. For the present study, the participants received a telephone call inviting them to sign an additional informed consent approved by the Bioethics Committee of the University of Pisa (n. 4/2018). They completed the validated Italian version of the Multidimensional Assessment of Interoceptive Awareness (MAIA) [19], and were submitted to hypnotic assessment through the Italian version of the Stanford Hypnotic Susceptibility Scale, form A [57]. Since both interoception [58] and hypnotizability [37,38] are stable in adulthood, MAIA and hypnotizability scores could be associated with earlier acquired EEG signals. Data were anonymized for further analysis. 

### 2.2. Experimental Procedure

#### 2.2.1. Signals Acquisition and Analysis

EEG and polygraphic recordings had been performed using a Micromed Morpheus device at home for an earlier study [56]. The participants were invited to go to bed at their usual time and sleep until waking spontaneously. Signals were acquired at a sampling rate of 512 Hz through 20 electrodes placed in accordance with the International 10–20 System, grounded at Pz, with Fpz as the reference electrode. In parallel, polygraphic traces were recorded, i.e., 1 ECG (bipolar derivation placed symmetrically around the sternum between the 3rd and 4th ribs), 2 EOG (left and right vertical), and 2 EMG derivations (electrodes placed on the chin over the suprahyoid muscles). Polysomnographic (PSG) data were exported in EDF+ format for further analysis. EEG and polygraphic signals were visually scored by a trained technician (MDG) according to the American Academy of Sleep Medicine Scoring Manual Updates for 2017 (Version 2.4) [59]. Epoch-by-epoch scoring was carried out on 30-s-long segments (Alice Sleepware software), and each epoch was assigned to the N1, N2 N3, or REM sleep stages, or to awake states [60]. 

For the present study, EEG signals were first pre-processed in EEGLAB [61]. Signals were band-pass filtered between 0.3 and 30 Hz and re-referenced to a common average reference, and noisy channels were interpolated. Independent component analysis (ICA) was then applied to identify and remove ICs reflecting eye movements, eye blinks, and cardiac artifacts, according to their topographic patterns. Signals were then epoched according to the ECG R-peak events and, after baseline correction (200 msec pre-R-peak), the HEP was computed separately for each sleep stage using ERPLAB [62]. In accordance with previous literature, the HEP was extracted by creating EEG epochs of 800 msec: from −200 ms to 600 ms, relative to the R peaks [55]. Sixty-four epochs were rejected from the analysis based on visual inspection, if the channels had a linear drift > 5 µV, or if the signal was larger than 2 standard deviations from the mean probability distribution of all epochs (mean rejection: N2, 21.9%; N3, 22.7%; REM, 19.4%). The number of analyzed epochs per subject (mean ± SD) was: N2, 4018 ± 1685; N3, 3714 ± 2276; REM, 3633 ± 1676. There were no significant differences in the number of analyzed epochs among the different stages (F(1,35) = 0.878, *p* = 0.38). The mean HEP amplitude for each electrode and sleep stage was then calculated in two different time windows: 200–350 ms—hereby defined as the early time interval—and 400–600 ms, defined as the late time interval. Fronto-central regions were chosen for analysis owing to their involvement in interoception in the general population [2,3,4,5,6,7,63], as well as in hypnotizability-related differences [52]. 

ECG signals were preprocessed using homemade MATLAB scripts to correct for impulsive noise, powerline interference (50 Hz), and baseline wandering. Impulsive noise is the sudden burst noise of short duration, caused mainly by electronic devices, and electrosurgical noise in biomedical signals at the time of acquisition. This noise was removed by applying a median filter (60 ms window) to each ECG channel and computing the absolute difference between the original and the median. A threshold value was computed based on their maximum absolute differences. The ECG signal in each interval where the absolute difference exceeded this threshold was replaced with the average of the signal before and after the interval. Powerline interference was removed by applying notch filters (forward–backward, zero phase, 1 Hz bandwidth) at 50 Hz or 60 Hz and its next three harmonics. Finally, to remove baseline wandering, for each channel we estimated a baseline signal by applying a low-pass first-order Butterworth filter (cutoff frequency at 5 Hz) in the forward and backward directions. The detrended signal was obtained as the difference between the original and baseline signals. 

ECG signals were interpolated to 1024 kHz via Fourier transform method to obtain a precise time location of the QRS complexes that were detected by applying a threshold to the absolute amplitude of a filtered derivative signal [64]. The series of the distances (msec) between consecutive R waves was calculated (tachogram) to measure the mean RR and the heart rate variability (HRV). HRV was measured in the frequency domain through the low-frequency (LF) and high-frequency (HF) components of the tachogram power spectrum. HF components are related to the short-term, parasympathetic variability, while LF components reflect the long-term, mainly—although not exclusively—sympathetic variability of the RR interval series [65]. In addition, HRV was studied in the time domain via the standard deviation of RR intervals (SDNN), reflecting the overall heart rate RR variability and [66], thus, also including the variability due to the baroreflex [67] and to the renin–angiotensin–aldosterone system [68], by the root-mean-square of successive differences in RR (RMSSD) related to parasympathetic control [69], and by the SDNN/RMSSD ratio [70].

#### 2.2.2. MAIA Questionnaire 

The MAIA questionnaire provides a measure of both interoceptive awareness and sensitivity [19,71,72]. It is a self-administered questionnaire consisting of 32 items, grouped into 8 scales: *noticing* (awareness of body sensations), *not distracting* (tendency to ignore sensations of pain/discomfort), *not worrying* (tendency to not experience emotional distress due to pain or discomfort), *attention regulation* (ability to focus attention to body sensation), *emotional awareness* (awareness of the relation between body sensations and emotional states), *self-regulation* (ability to regulate psychological distress by attention to body sensations), *body listening* (ability to listen to the body), and *trusting* (comfortable experience of one’s body). 

#### 2.2.3. Stanford Hypnotic Susceptibility Scale, form A (SHSS, A) 

SHSS, A is a behavioral scale [57] consisting of 12 items related to the respondent’s proneness to motor inhibition, dissociation, and hallucination. Each item may be passed (score = 1) or not passed (score = 0). SHSS, A classifies highly (highs, score ≥ 8 out of 12), moderately (mediums, score 5–7), and less hypnotizable individuals (lows, score ≤ 4). The Italian validated version of SHSS, A was used.

### 2.3. Statistical Analysis

The statistical package SPSS.20 was used for analyses. After normality assessment (Kolmogorov–Smirnov), non-parametric analyses (Wilcoxon), or repeated-measures ANOVAs were conducted between sleep stages on the autonomic state indices (RR, RMSSD, SDNN, SDNN/RMSSD, LF/HF, LF, and HF absolute power) for the entire sample, with the significance level set at *p* = 0.007 after Bonferroni correction. Repeated-measures ANOVAs were applied to the HEP amplitudes according to the following experimental design: 3 stages (N2, N3, REM) × 2 sides (left, right) × 2 areas (frontal, central). The Greenhouse–Geisser correction for non-sphericity was used when necessary, and contrast analyses were conducted between sleep stages. The significance level was set at *p* = 0.05. Hypnotizability (highs, mediums, lows) could not be used as a grouping variable, owing to the small number of highs and mediums enrolled in the sample.

For each sleep stage (N2, N3, REM), Spearman’s correlation coefficients between MAIA scores and HEP amplitudes (significance level set at *p* = 0.006 after Bonferroni correction), as well as between autonomic indices and HEP amplitudes, were computed (*p* = 0.007 after Bonferroni correction). Partial correlations between the same variables were also computed by removing the effects of hypnotizability, after assessing the absence of linear correlation between MAIA and hypnotizability scores.

## 3. Results

### 3.1. Preliminary Assessment

Total sleep time (mean ± SD, min; 453.59 ± 54.63), N2 (150.21 ± 52.22, min), N3 (142.17 ± 48.48, min), and REM (90.50 ± 29.15, min) stage durations, and MAIA scores (Table 1) were within their normal ranges [19,57,60]. 

The sample included 6 highs (SHSS score, 9.17 ± 0.75), 9 mediums (SHSS score, 6.11 ± 0.78), and 24 lows (SHSS score, 0.83 ± 1.13). The observed distribution of hypnotizability was not Gaussian, as most reported [73], which may be due to the small sample size and/or to the bias introduced by having enrolled only the participants who had previously consented to be equipped with PSG recordings [56]. MAIA and SHSS scores were not linearly correlated with one another. 

### 3.2. RR and HRV

RR (Figure 1a) showed a significant difference between sleep stages (χ^2^ = 12.44, *p* = 0.002, W = 0.141) sustained by a longer duration of N2 than of REM (Z = 3.56, *p* = 0.0001), and no significant differences between N2 and N3, or between N3 and REM. 

SDNN exhibited a significant stage effect (F(2,31) = 7.249, *p* = 0.007, η^2^ = 0.200), with higher values during N2 (mean ± sd (sec); 0.12 ± 0.11) than N3 (mean ± sd; 0.08 ± 0.05; F(1,31) = 5.769, *p* = 0.023), and during REM (mean +SD (s); 0.13 ± 0.12) than N3 (F(1,31) = 9.050, *p* = 0.005). No significant difference was observed between N2 and REM. 

RMSSD (mean ± SD (s); N2: 0.11 ± 0.08; N3: 0.09 + 0.06; REM: 0.11 ± 0.07) did not exhibit a significant stage effect. The SDNN/RMSSD ratio exhibited a significant stage effect (F(2,31) = 23.61, *p* = 0.0001, eta2 = 0.449). Contrast analysis revealed values significantly higher in REM (mean ± SD (s); 1.39 ± 0.32) than in N2 (mean ± SD (sec); 1.15 ± 0.29, F (1,31) = 18.91, *p* = 0.0001) and N3 (mean ± SD; 1.03 ± 0.17; F(1,31) = 44.03, *p* = 0.0001), and higher in N2 than in N3 (F(1,31) = 5.399, *p* = 0.027).

LF/HF (Figure 1b) was significantly different between sleep stages (χ^2^ = 13.00, *p* = 0.002, W = 0.221). It was higher during N2 (Z = 2.582, *p* = 0.010) and REM (Z = 3.366, *p* = 0.001) than during N3. In more detail, LF absolute power (Figure 1c) was not significantly different between stages, although contrast analysis revealed significantly higher values in N2 than in N3 (F (1,31) = 4.329, *p* = 0.044). HF absolute power (Figure 1d) differed between sleep stages (χ^2^ = 9.19, *p* = 0.012, W = 0.191), being higher in N2 (Z = 2.31, *p* = 0.012) and in N3 (Z = 2.71, *p* = 0.007) than in REM.

### 3.3. HEP, IS, and Hypnotizability

The amplitudes of early (200–350 ms) and late (400–600 ms) HEP components at frontal (F3, F4) and central sites (C3, C4) are shown in Figure 2. 

Across the entire sample, the amplitudes were pre-eminently negative, but this was due to the predominance of lows, as highs and mediums displayed pre-eminently positive values (Figure 3).

As reported in Table 2, the left hemisphere’s early HEP component tended to be significantly smaller than that of the right hemisphere (side effect). The significant stage x area interaction observed for the early HEP component was sustained by larger negativity in the frontal than in the central regions, and by larger frontal negativity in N3 than in N2 and REM, with no significant difference between N2 and REM (Table 2). The late HEP component displayed only a significant area effect, indicating larger HEP amplitudes at frontal than at central sites (Table 2).

No significant correlation between HEP amplitude and autonomic indices, nor between hypnotizability and HEP amplitude, was found at any sleep stage. Partial correlations removing the effects of hypnotizability did not disclose significant correlations between HEP amplitudes and autonomic indices. 

Associations were observed between HEP amplitudes and a few IS dimensions. During N2, in fact, we observed significant correlations between the amplitude of the later HEP component at central sites and *body listening* (ρ = 0.670, *p* = 0.0001) and *trusting* (ρ = −0.824, *p* = 0.0001) (Figure 4a). In N3, the amplitude of same central component correlated significantly with *self-regulation* (ρ = 0.670, *p* = 0.0001) and *trusting* (ρ = −0.606, *p* = 0.0001) (Figure 4b). During REM, the HEP amplitude correlated significantly with *self-regulation* (ρ = 0.824, *p* = 0.0001) (Figure 4c) and *body listening* (ρ = 0.606, *p* = 0.0001). All correlations between IS and HEP amplitude became non-significant after removing the effects of hypnotizability.

## 4. Discussion

The aims of this study were to investigate the association between interoceptive sensitivity and HEP amplitude during sleep, and to test whether hypnotizability moderated this association. We also replicated the earlier observed changes in the autonomic state and HEP amplitude occurring during sleep, and their lack of significant association (Section 4.1 and Section 4.2). Our results provide novel information supporting the relation between HEP amplitudes, interoceptive sensitivity, and hypnotizability during sleep (Section 4.3 and Section 4.4).

### 4.1. Autonomic and HEP Changes across Sleep Stages

The observed RR changes, consisting of decreases in RR from N2 to REM, but not of the expected increases from N2 to N3, may appear to be contrast with most reports [74,75]. It should be noted, in this respect, that the circadian cycle influences nocturnal heart rate, which exhibits the lowest values in the middle part of the night; that deep sleep is more frequent, longer, and accompanied by the lowest heart rate during the first half of the night; and that the present study includes data from all sleep cycles [76,77]. Notably, earlier studies on the same participants revealed longer N3 duration in mediums than in both highs and lows [59], which may have biased the present autonomic findings. 

The HRV of each sleep stage is also very variable through sleep cycles, and the reduction of the overall RR variability is the most reliable index of the shift from light to deep sleep [78], which is confirmed in the present study by the observed SDNN changes. 

The RR changes observed across sleep stages can be accounted for by spectral components, as LF decreased in the absence of significant increases in HF during N3. Moreover, LF did not change during REM with respect to N2 and N3, while HF decreased in REM with respect to both N2 and N3. In the time domain, the ratio SDNN/RMSSD is in line with the LF/HF changes and supports the view that the sympathetic–parasympathetic balance shifts towards a pre-eminently sympathetic control during REM [75,76]. 

### 4.2. HEP Amplitude Changes across Sleep Stages 

Pre-eminently positive and negative peak-to-peak HEP amplitudes have been described in participants with low and high interoceptive accuracy, respectively [79]. This can account for the mostly negative components of the HEP recorded in the present study, as most of the enrolled subjects were lows, who are characterized by a higher number of detected heartbeats compared to highs/mediums [54]. Preliminary results obtained in our sample have shown, in fact, positive HEP components in highs/mediums and negative components in lows, with significant differences between merged highs and mediums with respect to lows [80]. Since the larger the HEP amplitude, the higher interoceptive accuracy [81], the present study’s qualitative observations suggest a better cortical representation of the heartbeat in lows, which is consistent with behavioral measures of interoceptive accuracy [53]. Nevertheless, the association between HEP amplitude and behavioral measures of interoception is moderated by demographic, situational, and emotional variables [82], which should be considered in further research.

Findings show a greater amplitude of both early (200–350 ms) and late (400–600 ms) HEP components at frontal than at central sites across all sleep stages, as previously observed [2,3,4,5,6,7,35]. Larger frontal than central HEP amplitudes were also found, in fact, in awake conditions of enhanced arousal [62], emotion [22,23,25], and mood alteration [31].

The *quasi-significant* lower HEP amplitude at left than right hemispheric sites may be accounted for by the pre-eminent activity of the right insula, which is responsible for awareness of interoceptive signals, as measured by heartbeat-counting tasks [9,83]. 

In line with other studies [19,25,26], no linear correlation was found between autonomic variables and HEP amplitude, although the variability of autonomic-related cortical activities may be influenced by personality traits [25] not studied in the present research, and possibly buffering/counteracting the influence of the autonomic state. 

In the comparison between sleep stages, the frontal amplitudes of the early HEP component were larger in N3 than in N2 and REM, as earlier reported for the entire 200–600-ms HEP interval [32]. The similar HEP amplitudes observed during REM and N2 contrast with what occurs for the processing of exteroceptive information, which can be detected during REM only occasionally and when evoked by salient stimulation [35] and is consistent with the observation of similar responses to TMS administered during wakefulness and REM sleep [84]. A possible interpretation of this finding is that the interoceptive information is only partially dependent on the thalamic pathway as, in parallel with thalamic pathways, it is conveyed from the ventrolateral medulla, parabrachial nucleus, and periaqueductal gray to the hypothalamus and amygdala, reciprocally connected to the insula, cingulate, and frontal regions [1]. 

### 4.3. Association between IS, HEP Amplitude, and Hypnotizability

This is the first report of association between HEP amplitude during sleep and interoceptive sensitivity, which had been already observed during wakefulness in resting [11] and task conditions [27,85,86,87]. Their association during sleep (when attention is not focused on heartbeats [26], and vigilance is reduced) suggests that interoceptive sensitivity co-operates with interoceptive accuracy to the HEP amplitude. We may hypothesize that IA and IS represent a lower and higher level of interoceptive processing, respectively, according to the hierarchic model of interoception proposed by Critchley and Harrison [3]. 

The correlations we found regard the MAIA dimensions related to the attention to/confidence in the body, and the late HEP component representing the cognitive aspect of information processing [79]. The MAIA dimensions positively associated with the late HEP component indicate attention to body sensation (*self-regulation*, *attentional regulation*, *body listening*), which is consistent with the observation of greater HEP amplitudes in conditions of enhanced arousal [62], emotion [22,23,25], and mood alteration [31]. A negative correlation was found between HEP amplitudes and a comfortable experience of the body (*trusting*), in line with the observation of lower HEP amplitudes in patients with depersonalization [12].

The absence of significant linear correlations of hypnotizability with IS (MAIA scores) was expected, as earlier studies showed higher values of multiple MAIA scales in highs with respect to both mediums and lows [54]. Partial correlations revealed, however, that hypnotizability is greatly involved in the relation between HEP amplitude and IS dimensions, as removing the effects of hypnotizability abolishes those correlations. 

We speculate that the highs’ greater functional equivalence between imagery and perception [88]—also observed for interoceptive imagery [89]—would afford them an advantage in treatments aimed at modifying the experience and the interpretation of interoceptive signals.

### 4.4. Limitations and Conclusions

A limitation of the study is the small number of highs and mediums and the greater number of women with respect to men enrolled in the sample, together with the low effect size of a few comparisons conducted on the entire sample. This point could not be addressed because the study was a retrospective analysis of the EEG signals recorded in participants enrolled for an earlier study and not previously classified for hypnotizability. Further studies should enroll samples allowing for a direct comparison between highs, mediums, and lows by using hypnotizability as a grouping variable, rather than merely as a moderating factor. Moreover, the relation between HEP amplitude and interoceptive accuracy should be clarified. Moreover, cardiac signals are not exclusively interoceptive [28], and the heartbeat detection test may not be the best tool with which to assess interoceptive accuracy [18].

Despite the above limitations, however, the present study supports earlier findings describing the changes in HEP amplitude occurring during sleep in the general population [32] and confirms the absence of significant correlations between the sympathetic–parasympathetic state and the amplitude of the heartbeat-evoked cortical potential [12,26,63]. Its novelty consists of the report of associations between a few dimensions of interoceptive sensitivity—attention to the body, and confidence in it—and HEP amplitudes during sleep, and of the relevance of hypnotizability to these associations.

## Figures and Tables

**Figure 1 brainsci-11-01089-f001:**
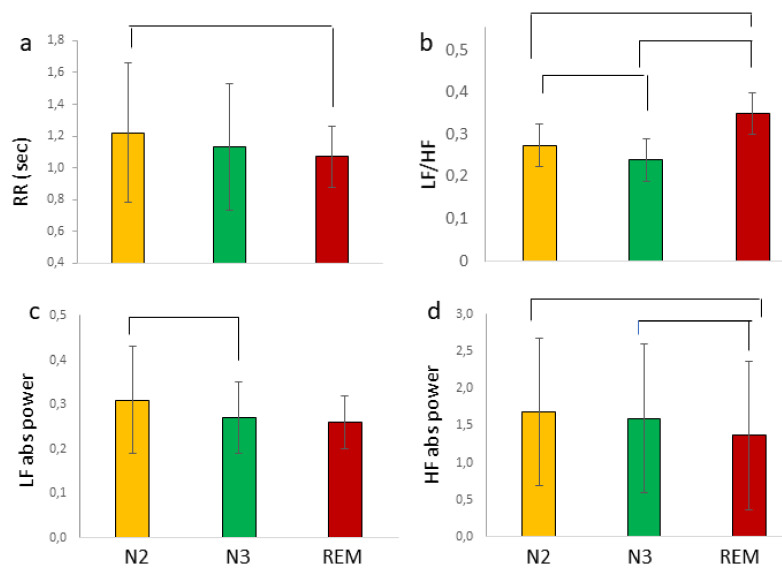
Autonomic states during sleep. Lines indicate significant differences between sleep stages for RR (**a**), LF/HF (**b**), LF absolute power (**c**), and HF (**d**) absolute power (mean, SD).

**Figure 2 brainsci-11-01089-f002:**
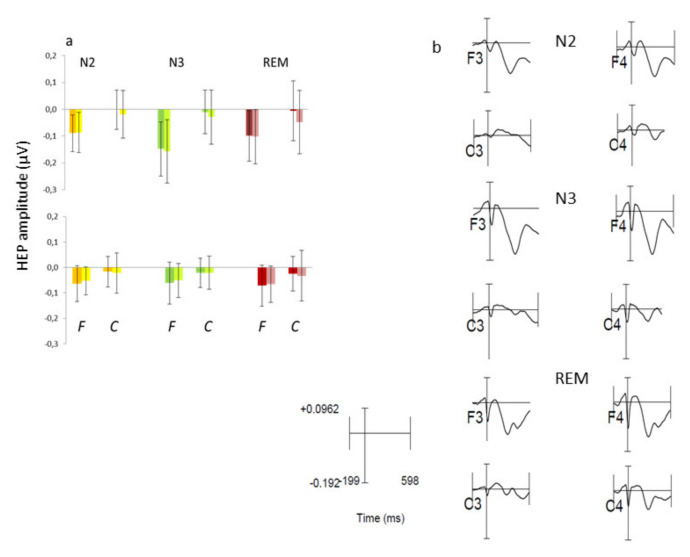
HEP amplitudes (mean, SD) during N2, N3, and REM sleep at frontal (F3, F4) and central (C3, C4) sites. (**a**) Dark and light colors indicate left and right sites, respectively, in N2 (yellow), N3 (green), and REM (brown). Upper and lower panels: early and late HEP components, respectively. (**b**) Average HEP signals at frontal and central sites.

**Figure 3 brainsci-11-01089-f003:**
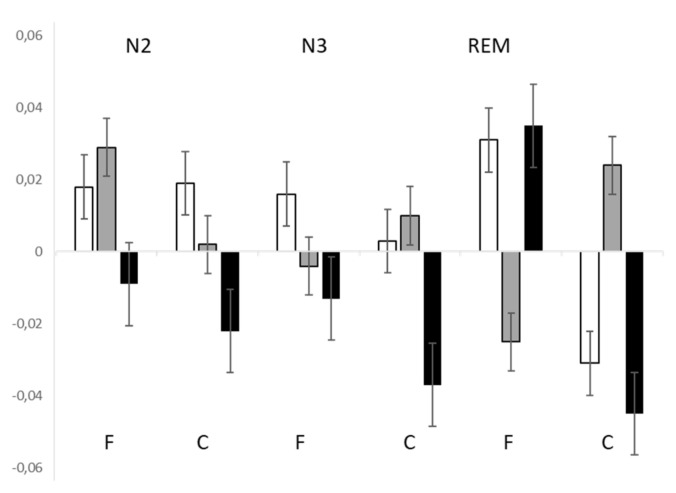
HEP amplitudes (mean, SD) in highs, mediums, and lows. White columns: highs; grey columns: mediums; black columns: lows. F: frontal sites C: central sites.

**Figure 4 brainsci-11-01089-f004:**
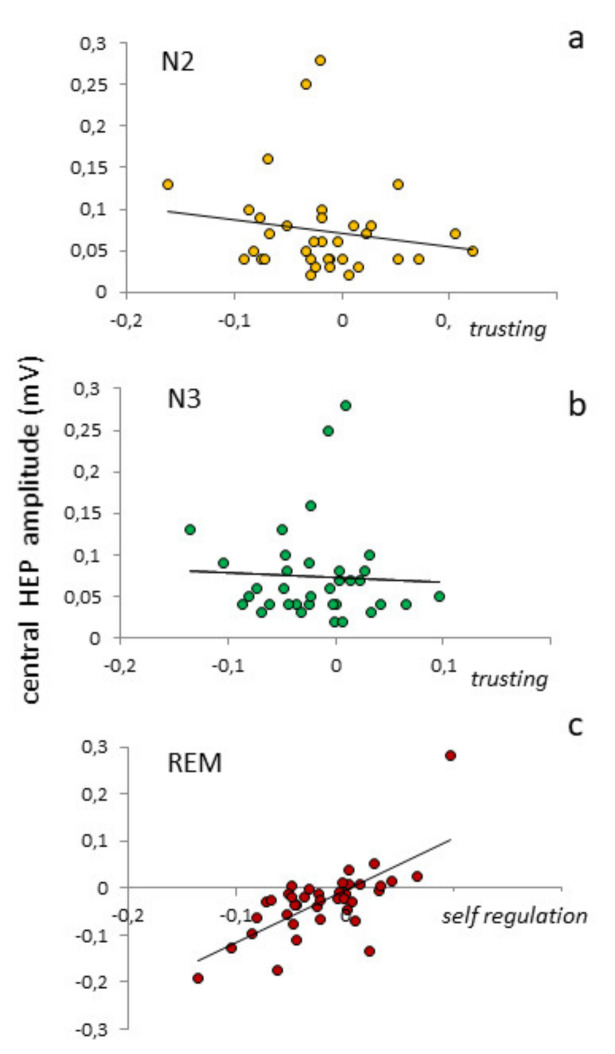
Correlations between MAIA dimensions and the amplitude of the late HEP component at central sites duing N2 (**a**), N3 (**b**) and REM (**c**).

**Table 1 brainsci-11-01089-t001:** MAIA scale mean values (SD).

Scale	Mean (SD)	Cronbach’s α
*noticing*	3.05 (0.99)	0.787
*not distracting*	2.49 (1.19)	0.645
*not worrying*	2.56 (0.94)	0.814
*attentional regulation*	2.66 (0.85)	0.981
*emotional awarenss*	0.90 (0.99)	0.783
*self regulation*	2.15 (1.16)	0.655
*body listening*	2.11 (1.29)	0.825
*trusting*	2.95 (1.00)	0.834

**Table 2 brainsci-11-01089-t002:** HEP amplitude; ANOVA.

	Effect	F	df	*p*	η^2^			
Early HEP								
	Stage	3.78	2,76	0.042	0.090			
						N2 < N3	F (1,38) = 17.61	*p* = 0.0001
						N2 = REM		
						N3 = REM		
	Side	3.91	1,38	0.055	0.093	left < right		
	Area	9.99	1,38	0.0001	0.723	F > C		
	Stage × Area	21.96	2,38	0.0001	0.366			
						N2, F > C	t = 8.81	*p* = 0.0001
						N3, F > C	t = 9.97	*p* = 0.0001
						REM. F > C	t = 7.43	*p* = 0.0001
					Frontal	N2 < N3	t = 5.24	*p* = 0.0001
						N3 > REM	t = 2.79	*p* = 0.008
						N2 = REM		
					Central		ns	
Late HEP								
	Area	13.87	1,38	0.001	0.267	F > C		

## Data Availability

The data underlying this article will be shared upon request.

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
