# Peer review of "Heartbeat-Evoked Cortical Potential during Sleep and Interoceptive Sensitivity: A Matter of Hypnotizability"

_brainsci, 2021, doi:10.3390/brainsci11081089_

Round 1
Reviewer 1 Report
The paper adresses interesting aspect. And the results are of importance in the field. However, several issues should be addressed:
Should be: Hearbeat evoked potential, and HEP – no need to use plural
A link from wakefulness studies to studies in sleep and relation to hypnosability is necessary to make in the abstract
Make sure in the introduction what is IS, and that it stands BOTH for awareness and sensitivity
Lines 37-38 – In the way it is presented I do not understand how this relates to the scope of present work, and to interoception per se. Please relate everything to autonomic activity in a more clear way. For example how it is related to hypnosability? How hypnosability is related to interoception? Please elaborate this further.
Lines 87-88 – reference should be added
Lines 93-94 how replication can be preliminary investigation?
Lines 127-135 reference is incorrect – number 56 stands for Eeglab, erplab and something else. This continues further on, f.e line 350. Please check.
Line 137: “a standard deviation > 2 standard deviations” makes no sense
Line 147: impulsive artefacts?
Lines 146-148 please make clear what exactly was made to the data? To “improve signal-to-noise ratio”?
Please make clear what version of MAIA was used – Italian, or English?
Please provide Chronbach’s alpha for the scales.
Line 197. What are the measures to express total sleep time? Minutes?
Line 198. Normality range – please rephrase
Line 2018 “consisting only of longer” does not make sense, please rephrase
Figure 2: captions do not correspond to Figure 2 – looks like there are mixed with Fig 3, but not sure – it is not possible to evaluate results in this way. It is also strange to put amplitude of evoked potentials in EEG in mV – I presume it should be microV
Formatting of tables
What is displayed as numbers and equations in Fig 4? Correlations were significant as reported in Results, but numbers in the Figure does not to be meaningful.
Line 364 “vigilance is reduced, suggests that interoceptive sensitivity co-operates with interoceptive accuracy to the HEPs amplitude” please rephrase
I would recommend language editing - some parts are difficult to follow
Author Response
We thank the reviewer for her/his helpful comments. We have addressed all of them. The line numbers may not correspond to the original version. All changes are in red characters
The paper adresses interesting aspect. And the results are of importance in the field. However, several issues should be addressed:
Should be: Hearbeat evoked potential, and HEP – no need to use plural
Corrected in the text, tables and figures
A link from wakefulness studies to studies in sleep and relation to hypnosability is necessary to make in the abstract
We have modified the abs
Make sure in the introduction what is IS, and that it stands BOTH for awareness and sensitivity
The MAIA questionnaire evaluates both aspects, although its earlier form was addressed to interoceptive sensitivity. We have clarified this in the text
Lines 37-38 – In the way it is presented I do not understand how this relates to the scope of present work, and to interoception per se. Please relate everything to autonomic activity in a clearer way. For example, how it is related to hypnosability? How hypnosability is related to interoception? Please elaborate this further.
We have included slight text modifications to clarify this aspect.
In a different part of the text (line 83) we have also mentioned that high hypnotizability is associated with higher parasympathetic tone during relaxation, reduced sympathetic activation in the shift from sitting to standing position and scarcely impaired release of endothelial nitric oxide during mntal stress and nociceptive stimulation. References have been added
lines 87-88 – reference should be added
done
Lines 93-94 how replication can be preliminary investigation?
done
Lines 127-135 reference is incorrect – number 56 stands for Eeglab, erplab and something else. This continues further on, f.e line 350. Please check.
corrected
Line 137: “a standard deviation > 2 standard deviations” makes no sense
The sentence was rephrased.
Line 147: impulsive artefacts?
We replaced “impulsive artifacts with “impulsive noise” and we better specified which is the source of this noise.
Lines 146-148 please make clear what exactly was made to the data? To “improve signal-to-noise ratio”?
The pre-processing steps applied to the ECG signals were better detailed. The phrase “to improve signal-to-noise ratio was removed since it was a misprint.
Please make clear what version of MAIA was used – Italian, or English?
This was reported in the original version
Please provide Chronbach’s alpha for the scales.
added
Line 197. What are the measures to express total sleep time? Minutes?
added
Line 198. Normality range – please rephrase
Rephrased
Line 2018 “consisting only of longer” does not make sense, please rephrase
Line 232? Substituted with “sustained by”
Figure 2: captions do not correspond to Figure 2 – looks like there are mixed with Fig 3, but not sure – it is not possible to evaluate results in this way. It is also strange to put amplitude of evoked potentials in EEG in mV – I presume it should be microV
Thanks for noticing that. We have corrected Figures and the Figure legend.
Formatting of tables
we provide Tables in word format
What is displayed as numbers and equations in Fig 4? Correlations were significant as reported in Results, but numbers in the Figure does not to be meaningful.
We have erased the line equations from the figure
Line 364 “vigilance is reduced, suggests that interoceptive sensitivity co-operates with interoceptive accuracy to the HEPs amplitude” please rephrase
Line 378? Rephrased
I would recommend language editing - some parts are difficult to follow
Done

Reviewer 2 Report
Billeci & colleagues use measures of interoception, hypnotisability and neurophysiology to determine EEG amplitudes ( HEPs amplitude) during sleep and its association with the autonomic state and IS, and to investigate the role of hypnotisability plays for this.
It is an extensive investigation with ambitious questions posed. Unfortunately, the methodological problems contained make this work very hard to interpret (details below).
MAJOR:
I ) Introduction
Overall this work is unfortunately motivated by previous work that used methods that have been shown to be invalid to measure what they claim to measure (details below).
1 A great part of the introduction is spent on citing evidence for measures of interoception referring for example, to evidence from the domain of cardiac interoception. I am sure the authors must be aware that there is considerable doubt about the validity of measures used in this field (i.e. heart beat counting task, self-report interoception measures). Thus, I think it has to be at least clearly acknowledged that a great deal of criticism of this work exists – I suggest looking into work by
Zamariola et al., 2018, 2019 (amongst others, detailed references below).
Zamariola, G., Frost, N., Van Oost, A., Corneille, O., & Luminet, O. (2019). Relationship between interoception and emotion regulation: New evidence from mixed methods. Journal of Affective Disorders
Zamariola, G., Maurage, P., Luminet, O., & Corneille, O. (2018). Interoceptive accuracy scores
from the heartbeat counting task are problematic: Evidence from simple bivariate correlations. Biological Psychology
Desmedt, O., Luminet, O., Corneille, O. (2018) The heartbeat counting task largely involves non-interoceptive processes: Evidence from both the original and an adapted counting task. Biological Psychology
2 It is not clear why hypnotisability would be important to modulate the relationship between measures on interoception during sleep. Please clarify the motivation in your intro.
The abstract is missing a summary of the method employed and its way too detailed about HEP and frankly, a bit confusing. Please add some clarification on the experimental paradigm, maker it clearer what was done, what the question was, how it was investigated and what was found without too many details.
Provide justification for the sample size: was a power analysis run?
The experimental design comes across as quite convoluted and I feel that the reader needs more help in understanding the wider implications of the study for our understanding interoception / sleep. Please add some additional discussion to support this.
Author Response
We thank the reviewers for their helpful comments. We have addressed all of them. The line numbers may not correspond to the original version. All changes are in red characters
Billeci & colleagues use measures of interoception, hypnotisability and neurophysiology to determine EEG amplitudes (HEPs amplitude) during sleep and its association with the autonomic state and IS, and to investigate the role of hypnotisability plays for this.
It is an extensive investigation with ambitious questions posed. Unfortunately, the methodological problems contained make this work very hard to interpret (details below).
MAJOR:
I ) Introduction
Overall this work is unfortunately motivated by previous work that used methods that have been shown to be invalid to measure what they claim to measure (details below).
1 A great part of the introduction is spent on citing evidence for measures of interoception referring for example, to evidence from the domain of cardiac interoception. I am sure the authors must be aware that there is considerable doubt about the validity of measures used in this field (i.e. heart beat counting task, self-report interoception measures). Thus, I think it has to be at least clearly acknowledged that a great deal of criticism of this work exists – I suggest looking into work by
Zamariola et al., 2018, 2019 (amongst others, detailed references below).
Zamariola, G., Frost, N., Van Oost, A., Corneille, O., & Luminet, O. (2019). Relationship between interoception and emotion regulation: New evidence from mixed methods. Journal of Affective Disorders
Zamariola, G., Maurage, P., Luminet, O., & Corneille, O. (2018). Interoceptive accuracy scores
from the heartbeat counting task are problematic: Evidence from simple bivariate correlations. Biological Psychology
Desmedt, O., Luminet, O., Corneille, O. (2018) The heartbeat counting task largely involves non-interoceptive processes: Evidence from both the original and an adapted counting task. Biological Psychology
We wish to remark that our study was aimed at measuring HEP. The heartbeat detection test was used in an earlier study. In the revised text we have added the limitations of the heartbeats detection task and included references.
It is not clear why hypnotisability would be important to modulate the relationship between measures on interoception during sleep. Please clarify the motivation in your intro.
Clarified. Interoception processing may differ according to hypnotizability owing to hypnotizability-related brain morphofunctional differences,,and such differences may appear also during sleep.
The abstract is missing a summary of the method employed and its way too detailed about HEP and frankly, a bit confusing. Please add some clarification on the experimental paradigm, maker it clearer what was done, what the question was, how it was investigated and what was found without too many details.
The abstract has been modified. Nonetheless these requests seem a bit contrasting between each other. We tried to respond to both by re-writing the abs
Provide justification for the sample size: was a power analysis run?
As we reported in the original version of the paper, the study is a retrospective analysis of previously acquired signals. Thus, we could not decide the number of participants to enroll. Nonetheless, in the Results section, the effect size of all comparisons is reported.
The experimental design comes across as quite convoluted and I feel that the reader needs more help in understanding the wider implications of the study for our understanding interoception / sleep. Please add some additional discussion to support this.
In the original version of the paper we wrote: We speculate that the highs’ greater functional equivalence between imagery and perception [86], observed also for interoceptive imagery [87], would advantage them in treatments aimed at reappraisal of the experience and interpretation of interoceptive signals.
This explains why our findings , regarding HEP as a correlate of hypnotizability, are relevant to sleep and interoception as well as to their relation.

Round 2
Reviewer 1 Report
Authors have addressed my comments.
Reviewer 2 Report
I don't think this is fit for publications. I don't think any further imporvements are required.